# ESSEN: Improving Evolution State Estimation for Temporal Networks using Von Neumann Entropy

**Qiyao Huang**[†], **Yingyue Zhang**[†], **Zhihong Zhang**[†*], **Edwin R. Hancock**[‡]

[†] School of Informatics, Xiamen University, China

[‡]Department of Computer Science, University of York, York YO10 5GH, UK.

{30920211154155, 30920221154261}@stu.xmu.edu.cn

zhihong@xmu.edu.cn, edwin.hancock@york.ac.uk

## Abstract

Temporal networks are widely used as abstract graph representations for real-world dynamic systems. Indeed, recognizing the network evolution states is crucial in understanding and analyzing temporal networks. For instance, social networks will generate the clustering and formation of tightly-knit groups or communities over time, relying on the triadic closure theory. However, the existing methods often struggle to account for the time-varying nature of these network structures, hindering their performance when applied to networks with complex evolution states. To mitigate this problem, we propose a novel framework called **ESSEN**, an **E**volution **S**tate**S** awar**E** **N**etwork, to measure temporal network evolution using von Neumann entropy and thermodynamic temperature. The developed framework utilizes a von Neumann entropy aware attention mechanism and network evolution state contrastive learning in the graph encoding. In addition, it employs a unique decoder the so-called Mixture of Thermodynamic Experts (MoTE) for decoding. ESSEN extracts local and global network evolution information using thermodynamic features and adaptively recognizes the network evolution states. Moreover, the proposed method is evaluated on link prediction tasks under both transductive and inductive settings, with the corresponding results demonstrating its effectiveness compared to various state-of-the-art baselines[1].

## 1 Introduction

Recently, graph representation learning has demonstrated excellent performance for various types of static graphs [7; 14; 28; 18]. Indeed, the success of static graph representation learning has led to a growing interest in continuous-time dynamic graph representation learning. Temporal network representation learning has emerged as an active research area focusing on learning low-dimensional representations that capture topological and temporal properties. However, learning effective representations is still a difficult task in many temporal networks, which are naturally generated in real-world systems such as social networks[12] and citation networks. The evolving nature of these networks poses a significant challenge for network analysis and modeling as the relationships between nodes and their properties evolve. The existing methods often struggle to account for the time-varying nature of these network structures, hindering their performance when applied to networks with complex time-evolving states. However, capturing the evolution states of temporal networks suffers from the following challenges: (1) Temporal networks have different types of evolution states, such as periodic, linear, or non-linear changes in their structure over time. Moreover, the evolving patterns can change at different stages in the evolution of the network. As

---

[*]Corresponding author.

[1]Code is available at https://github.com/QiyaoHuang/ESSEN

37th Conference on Neural Information Processing Systems (NeurIPS 2023).

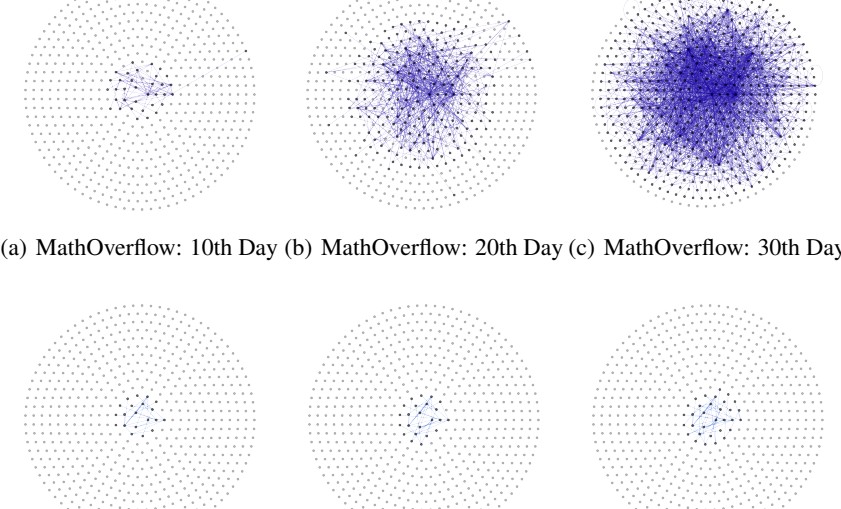

(a) MathOverflow: 10th Day  (b) MathOverflow: 20th Day  (c) MathOverflow: 30th Day

(d) BitcoinOTC: 10th Day    (e) BitcoinOTC: 20th Day    (f) BitcoinOTC: 30th Day

Figure 1: Network snapshots of the MathOverflow website and BitcoinOTC trading platform on the 10th, 20th, and 30th day. The black nodes represent the users who have connections, and the grey nodes represent the users with no edges prior to the snapshot time.

illustrated in Fig. 1, the rate of evolution varies at different times in both datasets. The MathOverflow network evolves rapidly and has obvious central nodes, which means meaningful topics will receive long-term attention. By contrast, in the BitcoinOTC network, early active users may quickly become dormant. This phenomenon requires algorithms that can perform extensive and effective recognition of diverse evolution states. (2) As time passes, temporal networks tend to accumulate more nodes and edges, resulting in an increasing number of possible connections and a rapidly growing neighborhood size for each node. This growth in the neighborhood size can lead to significant computational challenges when analyzing and modeling evolving patterns. Besides, many connections can quickly make structure recognition computationally intractable in large and complex networks, especially those recent methods based on the anonymous walk [24; 6; 20]. The time complexity is tightly related to the length and number of paths, so it is challenging to balance time consumption and algorithm effectiveness.

To overcome these shortcomings, in this paper, we use the von Neumann network entropy to improve evolution state estimation. Network entropy is a macroscopic representation of network structures widely used to characterize the salient features of static and dynamic network systems in biology, physics, and social sciences. In particular, von Neumann entropy has been successfully used to describe the structural properties of random, small-world, and scale-free networks [1; 2], and thus plays a crucial role in understanding the structural and topological complexity of network systems. Moreover, the ability of von Neumann entropy to capture structural information content aligns well with the time-evolving nature of temporal networks. This potentially allows us to adopt an information-theoretic perspective to enhance our understanding of the laws of the network's evolution and their capacity to both transmit and store information over time.

Unfortunately, computing the required network entropies for temporal networks can be computationally burdensome due to the required spectral decomposition. Hence, we approximate the von Neumann entropy to render it tractable in the context of temporal networks. Compared with alternative graph entropies, by approximating von Neumann entropy with low time complexity we can better adapt to the constantly evolving nature of complex temporal networks. Moreover, we can compute an approximate thermodynamic temperature for the temporal network, which provides an important way to monitor changes in network structure with time. Measuring the thermodynamic temperature and the von Neumann entropy provides a better understanding of the state of network evolution with time.

On this basis, we propose an **E**volution **S**tate**S** awar**E** **N**etwork (**ESSEN**). As shown in Fig. 2, ESSEN encodes node embeddings with state evolution information by utilizing two proposed techniques:

a) a von Neumann entropy aware attention mechanism and b) virtual evolution node representation learning. In addition, ESSEN employs a unique mixture of thermodynamic experts (MoTE) for the purposes of decoding. Specifically, we project the global von Neumann network entropy into each edge. The proposed von Neumann entropy aware attention mechanism aggregates over neighborhoods in both a virtual evolution graph and the original graph based on the von Neumann projected edge entropy. In the virtual evolution graph, future edges are hypothesized based on a test pertaining to future instants in time. The MoTE decoder evaluates the evolution state based on both the thermodynamic temperature and von Neumann entropy over both graph representations, providing a combined result from multiple experts. The decoder adaptively recognizes the network under various evolution states. Our framework is evaluated on transductive and inductive link prediction tasks. The experimental results demonstrate our method's effectiveness compared to various state-of-the-art baselines. The overall contributions of our work are summarized as follows:

- To our best knowledge, we are the first to utilize the von Neumann entropy in temporal network representation learning. We provide a method to expand the approximate von Neumann Entropy and approximate thermodynamic temperature to temporal networks.

- We propose a novel framework, namely ESSEN. The model introduces a new perspective to encode evolution-aware node representations using the von Neumann entropy aware attention mechanism and virtual evolution node representation learning. Furthermore, the model uses a novel decoder MoTE that adaptively recognizes temporal network evolution states.

- We evaluate our framework on link prediction tasks with transductive and inductive settings. The results show the effectiveness of our proposed method compared to various state-of-the-art baselines.

## 2 Preliminaries

**Temporal Network.** Formally, the temporal network can be denoted as $G = (V, E, T)$, where $V$ represents the set of nodes, $E \subseteq V \times V$ represents the set of links, and $T$ represents the set of timestamps. Each link $(u, v, t)$ signifies a connection between node $u$ and node $v$ at time $t$. The temporal network evolves over time, with links appearing at different timestamps. The temporal networks can also include attributes associated with nodes or links, providing further information about the entities or their interactions at specific timestamps.

**Dynamic Link Prediction.** In a temporal network $G = (V, E, T)$, the dynamic link prediction task aims to predict the presence or absence of a link at a future timestamp based on the observed network evolution history. Given a time window $T_w \subseteq T$, which contains the observed link data, denoted as $E_{T_w} \subseteq E$, the goal is to learn a function $f : (V, E_{T_w}, T_w) \to \{0, 1\}$ that assigns a probability score to the existence of a link $(u, v)$ at the future timestamp $t$. Mathematically, the function $f$ can be defined as:

$$f(u, v, t) = P(u, v | t, E_{T_w}),  \tag{1}$$

where $P((u, v)|t, E_{T_w})$ represents the probability of the link $(u, v)$ being present at the future timestamp $t$ given the observed network and the link data $E_{T_w}$ within the time window $T_w$.

**Evolution State.** Evolution state denotes the specific arrangements of nodes and edges at specific instants of time. These states can be characterized by a network topology, reflecting the evolving nature of the network over time. For example, social networks at specific evolution states will generate the clustering and formation of tightly-knit groups or communities over time via the tri-adic closure theory [29]. The theory is formally defined as $\exists u, v, w, w' \in V : (u, v), (v, w) \in E_{T_w}, (u, w), (u, w'), (v, w') \notin E_{T_w} \mapsto P(u, w | E_{T_w}) > P(u, w' | E_{T_w})$. Therefore, analyzing evolution states helps in understanding the temporal behavior of the network, identifying recurring patterns, predicting future states, and studying the impact of temporal dynamics on network properties and phenomena. In this paper, we aim to capture evolution states using thermodynamic entropy.

**Von Neumann Entropy.** Von Neumann Entropy is the quantum counterpart of the Shannon entropy. For a quantum system with a density matrix $\rho$ the von Neumann entropy is $S_{VN} = -\operatorname{Tr}(\rho \log \rho)$. The density matrix $\rho$ describes a system whose state is a mixture of pure quantum states $|\psi_i\rangle$, each with probability $p_i$, and is defined as $\rho = \sum_{i=1}^{T} p_i |\psi_i\rangle \langle \psi_i|$, where $T$ is the number of pure states. When defined in this way, the density matrix is Hermitian, i.e., $\rho = \rho\dagger$ and $\rho \geq 0$, $\operatorname{Tr}[\rho] = 1$, where $\dagger$

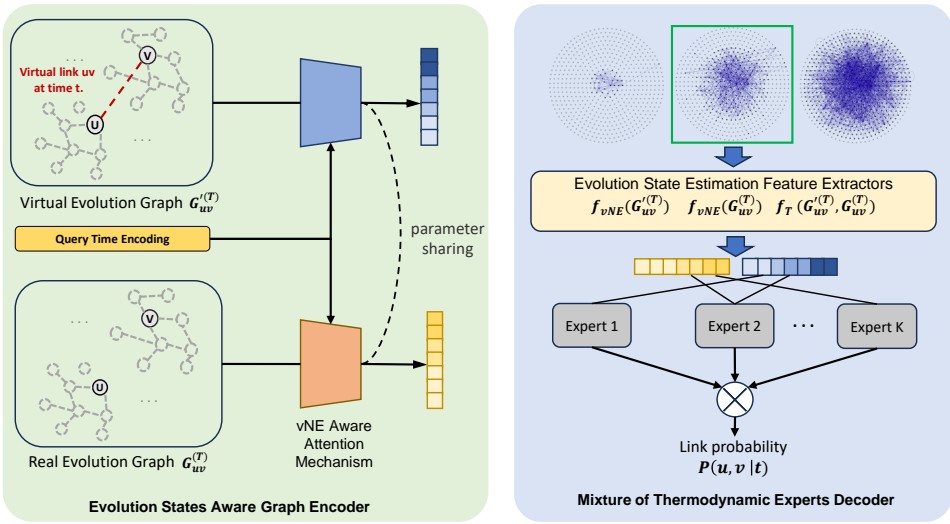

Figure 2: The framework of ESSEN. It comprises two modules: (1) Evolution States Aware Graph Encoder. (2) Mixture of Thermodynamic Experts Decoder.

represents conjugate transpose. It plays an important role in the quantum observation process, which can be used to calculate the expectation value of measurable quantities.

Severini et al. [17] show that a density matrix for a graph or network can be obtained by scaling the combinatorial Laplacian matrix $\tilde{L} = D - A$ (where $A$ is the adjacency matrix and $D$ is the diagonal degree matrix) by the reciprocal of the number of nodes in the graph $\mathcal{V}|$, i.e. $\rho = \frac{\tilde{L}}{|\mathcal{V}|}$. The interpretation of the scaled normalized Laplacian as a density operator opens up the possibility of characterizing a graph using the von Neumann entropy. With the definition of the density matrix adopted, the von Neumann entropy[17] can be computed from the normalized Laplacian spectrum as follows:

$$S_{VN}(G) = -\operatorname{Tr}(\rho \log \rho) = -\sum_{i=1}^{|\mathcal{V}|} \frac{\lambda_i}{|\mathcal{V}|} \log \frac{\lambda_i}{|\mathcal{V}|}, \tag{2}$$

where $\lambda_1, \ldots, \lambda_{|\mathcal{V}|}$ are the eigenvalues combinatorial Laplacian matrix. This form of von Neumann entropy has been shown to be effective for static network characterization.

## 3 The Proposed Method

### 3.1 Evolution State Estimation

**Von Neumann Entropy Computation in Temporal Network.** The von Neumann entropy can be computed in the static graph by Eq. 2. However, the application to temporal networks has two challenges: (1) The dynamic nature of the temporal network. Unlike static networks, where the structure remains constant, temporal networks capture time-evolving relationships and interactions. This dynamic nature introduces challenges in analyzing and modeling the network behavior, as the network topology and connectivity patterns may vary at different times. (2) The computationally expensive time complexity of obtaining the Laplacian eigenvalues. In Eq. 2, computing the eigenvalues of the Laplacian matrix is a computationally intensive task, and the time complexity is cubic in the number of nodes. In temporal networks, where the network structure changes over time, repeatedly calculating the Laplacian eigenvalues can become prohibitively expensive.

To address these challenges, we must simplify the network and efficiently approximate thermodynamic quantities. **First**, we select a specific time interval for the temporal network and aggregate edge weights or frequencies over this interval. The number of occurrences within the chosen time frame

determines the strength of an edge. Following this process, the temporal network can be projected to the time-independent 2-D plane, which provides a simplified representation of the underlying network structure at a specific time. Moreover, following [27], we use the approximate expression for the von Neumann entropy and reduce the computation to be quadratic in the number of nodes. The approximate von Neumann entropy is

$$S_{VN}(G_t) = 1 - \frac{1}{|V|} - \frac{1}{|V|^2} \sum_{(u,v) \in E} \frac{1}{d_u d_v}, \tag{3}$$

where $V$ is the node set of the temporal network, and $d_u$, $d_v$ are the degree of node $u$ and $v$ at time $t$. This approximation allows the von Neumann entropy to be computed without explicitly solving the eigensystem for the normalized Laplacian. Thus, the von Neumann entropy can be computed in quadratic time using the node degrees for pairs of nodes connected by edges. We introduce the details of the approximate method in Appendix A.

**Thermodynamic Temperature.** In thermodynamics, the thermodynamic state of a system can be fully described by an appropriate set of thermodynamic variables. We treat the von Neumann entropy as a thermodynamic network entropy and use the thermodynamic temperature to $\mathcal{T}$ measure fluctuations in network structure with time. Specifically, suppose that the graphs $G_1$ and $G_2$ represent the structure of a time-varying system at two consecutive epochs $t_1$ and $t_2$ respectively. For a thermodynamic system with a fixed number of particles, the change in energy is $de = \mathcal{T}dS - P_r dV_o$, where $P_r$ and $V_o$ denote the pressure and volume of the system respectively. It is important to stress that this equation holds and is valid for both reversible and irreversible processes for a closed system, since $e, \mathcal{T}, S, P_r$ and $V_o$ are all state functions and are independent of thermodynamic path. We assume the network evolves from time $t_1$ to time $t_2$ under constant-volume (isochoric process) as the time interval becomes small, i.e. $\Delta t \rightarrow 0$. As a result, for the path from $G_1$ to $G_2$ we have $dV_o = 0$ and $de = \mathcal{T}dS$. The reciprocal of the temperature $\mathcal{T}$ is the rate of change of entropy with average energy, i.e.,

$$\mathcal{T}(G_1, G_2) = \frac{de}{dS} = \frac{e(G_1) - e(G_2)}{S(G_1) - S(G_2)}, \tag{4}$$

where $S(G_1)$ is the graph entropy for the graph $G_1$ and $e(G_1)$ is the graph average energy. Moreover, the thermodynamic temperature can also be approximated when we represent the temporal network as a directed graph. The approximate computation method uses a low-order Taylor series that can be computed using the traces of powers of the normalized Laplacian matrix, avoiding explicit computation of the normalized Laplacian spectrum [26]. In summary, the temperature associated with the evolutionary transition between the two networks can be approximated as

$$\mathcal{T}(G_1, G_2) = -\frac{2}{k} + \frac{2}{3k} \cdot \frac{\mathcal{K}(G_1) - \mathcal{K}(G_2)}{\mathcal{J}(G_1) - \mathcal{J}(G_2)}, \tag{5}$$

where

$$\mathcal{J}(G) = \sum_{u,v \in V} \frac{A_{uv}}{d_u d_v}, \tag{6}$$

$$\mathcal{K}(G) = \sum_{u,v,w \in V} \frac{A_{uv} A_{vw} A_{wu}}{d_u d_v d_w}, \tag{7}$$

where $k$ is the Boltzmann constant, and $A$ is the adjacency matrix of the network. $\mathcal{J}(G)$ and $\mathcal{K}(G)$ are network structure statistics that can be interpreted as the probabilities of a random walker traversing specific edges or cycles in the graph. The computation of temperature is quadratic in the number of nodes.

**In summary**, by simplifying the network representation and using efficient approximations, the von Neumann entropy and the thermodynamic temperature can be computed effectively in temporal networks. These measures provide insights into the evolving nature of the network and enable the estimation of its evolution state.

## 3.2 Evolution States Aware Graph Encoder

**Von Neumann Entropy Aware Attention Mechanism.** In graph encoding, we utilize the von Neumann entropy to explore a more diverse and balanced distribution of attention weights with the

attention mechanism in the input neighborhood. This strategy helps us to learn network evolution states adaptively. According to Eq. 3, the global network entropy is a sum of contributions from individual edges. The von Neumann entropy of the edge connecting nodes $u$ and $v$ is

$$S_{VN}^{uv}(G_t) = \frac{1}{|E|} - \frac{1}{|V||E|} - \frac{1}{|E||V|^2}\frac{1}{d_u d_v}, \tag{8}$$

To encode entropy features into attention layers, we use the von Neumann edge entropy as a bias term in an attention module[22]. Moreover, we use a time position encoding module[25] to supplement the continuous time information for edges using the simplified von Neumann entropy. For a target node $u$ at time $t$, the attention weight $\alpha_v^{(l)}$ for the neighboring node $v$ in the $l^{th}$ layer is

$$\alpha_v^{(l)} = \frac{Q_u^{(l)}\left(K_v^{(l)}\right)^T}{\sqrt{d_n}} + S_{VN}^{uv}(G_t), \tag{9}$$

$$Q_u^{(l)} = (h_u^{(l-1)}\|e_0\|\phi(0))W_Q, \tag{10}$$

$$K_v^{(l)} = M_{v,t}^{(l)}W_K, \tag{11}$$

$$M_{v,t}^{(l)} = \left(h_v^{(l-1)}\|e_{uv,t}\|\phi(t_q - t)\right)W_M, \tag{12}$$

where "$\|$" is the concatenation operation. $W_K \in \mathbb{R}^{(d_n+d_t+d_e)\times d_n}$ and $W_Q \in \mathbb{R}^{(d_n+d_t+d_e)\times d_n}$ are the projection matrices to obtain the query matrices and key matrices, where $d_n$, $d_t$ and $d_e$ are the dimensions of the node representation, the time code, and the entropy bias, respectively. $e_0$ is an all-zero vector to keep the same dimension as $K$ and $V$, and $\phi(*)$ is the generic time position encoding module from [25], which encodes the difference between the edge timestamp and the query timestamp. $M_{v,t}^{(l)}$ is the message representation at time $t$ from node $v$ to $u$, where $h_v^{(l-1)}$ is node $v$'s hidden representation on the $(l-1)^{th}$ layer, $e_{uv,t} \in \mathbb{R}^{d_e}$ is the edge feature, and $t_q$ is the query time. Next, the model combines values with the attention weight aware of generating hidden representation $z_u^{(l)}(t)$ for node $u$. Finally, an MLP is used to combine the node representation of the previous layer with the neighborhood information:

$$h_u^{(l)} = MLP(h_u^{(l-1)}\|z_u^{(l)}(t)), \tag{13}$$

$$z_u^{(l)}(t) = \sum_{v\in\mathcal{N}_u} \text{softmax}_v\left(\alpha_v(t)\right)V_v(t), \tag{14}$$

$$V_v^{(l)} = M_{uv,t}^{(l)}W_V, \tag{15}$$

where $V_v^{(l)}$ is the value vector of neighbor node $v$, and $\mathcal{N}_u$ is a neighbor node set that connects with node $u$ before time $t$.

**Virtual Evolution Node Representation Learning.** Temporal networks follow time-dependent evolution laws. The emergence of nodes and edges at different times can often be predicted, i.e., future network states can be predicted from their past states and together with the evolutionary laws. Virtual evolution node representation learning utilizes historical evolution information together with the future potential evolution path to generate virtual node representations. Specifically, the dynamic link prediction task aims to predict the probability of the link between two nodes appearing at a future instant in time. We assume the link has been generated at the query instant and further construct a virtual evolution graph belonging to the query node pair based on this assumption. For example, given a node pair $(u, v)$ and the query time $t$, there is a virtual edge at time $t$ that connects the nodes $u$ and $v$ in the virtual evolution graph $G'_{uv}$. The approach makes the two node neighborhoods interconnected. We denote as $h'_u$ and $h'_v$ the virtual future node embeddings of nodes $u$ and $v$. Moreover, the thermodynamic temperature of the virtual evolution path can be computed because the instantaneous virtual evolution process is also an isochoric process we introduced in Section 3.1. Based on this, the decoding process can estimate the evolution states more comprehensively.

### 3.3 Mixture of Thermodynamic Experts Decoder

**Evolution State Feature Extractor.** We use the von Neuman entropy of the original graph, the von Neuman entropy of the virtual evolution graph, and the thermodynamic temperature between the two networks to construct a unique vector for evolution state estimation. The vector represents the network evolution states in a 3-D thermodynamic space, which is spanned by the von Neumann entropy at the two instants in time together with the thermodynamic temperature. The approximate von Neumann graph entropy for the original and virtual evolution graphs are computed by Eq. 3, and the approximate thermodynamic temperature is computed by Eq. 5. Furthermore, to control evolution-aware time intervals, which is important for large temporal networks, we compute these approximate thermodynamic quantities in the node neighborhood and set the sampled neighborhood size $N$.

**Expert Decoding Process.** The mixture of thermodynamic experts decoder dynamically selects the appropriate thermodynamic expert model based on the input expert assessment feature vectors. For each expert, we use a two-layer MLP model to represent. Then the MoTE decoder combines the output embedding of each expert model using respective expert weights to produce the final target score as follows:

$$score(u, v, t) = \sum_{i=1}^{Y} \sigma(W_i(h_u, h_v, h'_v - h_v, h'_u - h_u))\pi_i, \tag{16}$$

$$\pi_i = softmax_i((\mathcal{T}(G, G'_{uv})\|S_{VN}(G)\|S_{VN}(G'_{uv}))W_\pi), \tag{17}$$

where $Y$ is the total number of experts, $\pi_i$ is the mixing coefficient of expert $i$. Additionally, $W_i \in \mathbb{R}^{4d \times 1}$ is the weight matrices for expert $i$ and $W_\pi \in \mathbb{R}^{3 \times Y}$ is the weight matrix of the gate unit. Finally, $h_v$ and $h'_v$ are the embeddings of node $v$ in the original graph and the virtual time-evolved graph generated by the encoder.

### 3.4 Optimization

During training, we evaluated the convergence behavior of our model by monitoring the training and validation loss, ensuring that the model was not underfitting or overfitting. The loss function is:

$$\ell = \sum_{(v_i, v_j, t_{ij}) \in \mathcal{E}} \log P(v_i, v_j \mid t_{ij}) - \mathcal{Q} \cdot \mathbf{E}_{\tilde{v} \sim P(\tilde{v})} \log P(v_i, \tilde{v} \mid t_{ij}), \tag{18}$$

where $(v_i, v_j, t_{ij})$ is an observed edge on the temporal network, $\mathcal{Q}$ denotes the number of negative samples, and $P(\tilde{v})$ is the negative sampling distribution over the node space $\mathbf{E}$.

### 3.5 Computational Complexity Analysis

This section highlights the efficiency of our approach in calculating approximate thermodynamic quantities of temporal networks. Based on Eq.3 and Eq. 5, the time complexity of computing approximate von Neumann entropy and the approximate temperature is $O(|V|^2)$, where $|V|$ is the number of nodes in the network. Moreover, we compute the approximate thermodynamic quantities in the neighborhood for the large networks and set the sampled neighborhood size $N$. The computational complexity can be reduced to $O(N^2)$ in this setting. Therefore, the time complexity demonstrates scalability and establishes the feasibility of our method for moderate or large networks. Since the time complexity is controllable this means we can achieve efficient computation.

## 4 Experiments

### 4.1 Experimental Setup

**Datasets.** The temporal network datasets used in our experiments are divided into three categories: (a) QA: The "answers to questions"

Table 2: Statistics of the datasets.

| Dataset | Nodes | Edges | Timespan |
|---|---|---|---|
| MathOverflow | 21,688 | 107,581 | 2350 days |
| BitcoinOTC | 5,881 | 35,592 | 1903 days |
| BitcoinAlpha | 3,783 | 24,186 | 1901 days |
| Wikipedia | 9,227 | 157,474 | 30 days |

Table 1: Performance of AUC(%) for link prediction. The best results in each column are highlighted in bold font and the second-best results are underlined. We report the AP results in Appendix B.

| Task | Methods | MathOverflow | BitcoinAlpha | BitcoinOTC | Wikipedia |
|---|---|---|---|---|---|
| Transductive | JODIE | 86.07 ±0.48 | 91.14 ±0.18 | 92.29 ±0.11 | 93.58 ±2.00 |
| | DyRep | 80.77 ±0.65 | 79.39 ±3.17 | 79.21 ±4.10 | 94.22 ±0.27 |
| | TGN | 80.47 ±3.24 | 86.71 ±1.00 | 86.78 ±2.29 | 98.46 ±0.10 |
| | TGAT | 71.80 ±0.91 | 78.99 ±0.50 | 79.53 ±0.67 | 95.34 ±0.10 |
| | CAW | 53.82 ±0.28 | 64.70 ±0.93 | 73.95 ±1.22 | 98.96 ±0.10 |
| | TDLG | 84.02 ±0.16 | 92.83 ±0.22 | 93.48 ±0.22 | 88.93 ±0.09 |
| | NeurTWs | 92.56 ±0.51 | 93.95 ±0.41 | 95.75 ±0.01 | 94.54 ±0.87 |
| | ESSEN | **98.60 ±0.40** | **99.10 ±0.16** | **98.88 ±0.42** | **99.03 ±0.33** |
| Inductive | JODIE | 67.06 ±0.42 | 74.47 ±0.16 | 76.21 ±0.47 | 91.44 ±1.99 |
| | DyRep | 63.50 ±0.66 | 66.27 ±0.73 | 65.09 ±0.86 | 91.03 ±0.34 |
| | TGN | 64.50 ±1.17 | 69.36 ±0.94 | 76.52 ±1.25 | 97.70 ±0.18 |
| | TGAT | 60.02 ±0.75 | 66.42 ±1.17 | 66.62 ±1.99 | 93.99 ±0.30 |
| | CAW | 57.67 ±0.33 | 64.38 ±1.01 | 72.99 ±0.46 | 98.75 ±0.14 |
| | TDLG | 74.31 ±1.58 | 83.85 ±1.65 | 85.22 ±3.89 | 45.77 ±3.06 |
| | NeurTWs | 91.83 ±0.13 | 94.20 ±0.26 | 96.08 ±0.38 | 94.63 ±0.47 |
| | ESSEN | **98.33 ±0.28** | **98.07 ±0.64** | **98.67 ±0.31** | **98.80 ±0.10** |

dataset of MathOverflow. (b) Bitcoin trading data: BitcoinAlpha Dataset and BitcoinOTC Dataset [9; 8]. (c) Social networks: Wikipedia Dataset[10]. Table 2 gives more details concerning the properties of these datasets.

**Baselines.** In addition to reporting the performance of our ESSEN method, we report results for several popular dynamic graph learning methods, namely a) JODIE [10]; b) DyRep [21]; c) TGAT [25]; d) TGN [19]; e) CAW [24]; f) TDLG[4]; and g) NeurTWs[6]. We give more details concerning these baselines in Appendix C.

**Link Prediction Task Settings.** We evaluate our model on the link prediction task with two different settings:

- **Transductive Setting.** The model under the transductive setting is trained on the available nodes and their connections to predict future links between the nodes. The setting assumes that the network will not add unseen nodes at future testing times. This mainly evaluates the transductive ability of the model.

- **Inductive Setting.** The inductive setting predicts missing links for existing nodes together with potential new nodes that may be added at future times. This generalizes link prediction beyond the known nodes, considering the possibility of the addition of new nodes. In this way, it learns network patterns and characteristics to make predictions applicable to both known and unknown nodes.

**Implementation Training Details.** For each dataset, we used the training time points $T_{tr} = 70\%$ to split the dataset results in approximately 70%-15%-15% of the total edges [25]. The principal hyperparameters are set as follows: a) the number of attention heads $\mathcal{U} = \{2, 3\}$, b) the number of the GNN layers $\mathcal{L} = 2$, c) the maximum number of aggregated neighbors $n \in \{60, 80, 100\}$, d) the total number of experts in MoTE $Y = \{4, 6, 8, 10\}$, and e) the dimension of the node embedding $d_n = 172$. We use the ADAM optimization algorithm for model training with a learning rate 1e-3 and batch size of 128. All the models are implemented in PyTorch and evaluated on a single Tesla A100 GPU.

## 4.2 Results and Discussion

Table 1 reports the transductive and inductive link prediction task results on four datasets, demonstrating the state-of-the-art performance of our method on link prediction tasks. Our method significantly outperforms all baselines on all datasets. In particular, in the MathOverflow dataset, compared with NeurTWs, the second strongest baseline, ESSEN improved the AUC(%) by 5.04% and 6.50% on average in the transductive and inductive setting. The results demonstrate that our method has a

clear advantage for temporal networks. Specifically, our method performs well on both long and short-evolution time networks, while the effectiveness of baseline models varies significantly. CAW and TGAT have significant gaps in performance with MathOverflow and Wikipedia for all tasks. This indicates that our framework better represents networks with ever-changing evolution states. This superiority in performance can be attributed to a) our von Neumann entropy aware mechanism, b) the virtual evolution node representation learning, and c) the MoTE decoder. In addition, our method is effective under both transductive and inductive settings. By contrast, the baseline JODIE method cannot predict interactions well between unseen nodes because it pays more attention to node identities rather than the evolution states of temporal networks.

### 4.3 Ablation Study

To validate the effectiveness of the novel elements comprising ESSEN, we conduct a series of ablation studies and report the AUC results. We investigate the performance of the proposed modules with three ablated models on the MathOverflow and BitcoinAlpha dataset: a) ESSEN-$E$, here we remove the von Neumann edge entropy bias in the attention mechanism of ESSEN. b) ESSEN-$V$, here we remove the virtual evolution node representation and only use the node embeddings of the original graph for decoding. c) ESSEN-$D$, here we replace the MoTE decoder with a simple MLP decoder. In Fig. 3, we can see that the performance degrades without considering the von

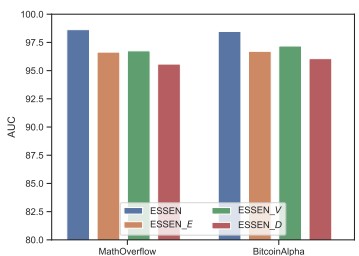

Figure 3: Ablation Study

Neumann entropy information for edges. This demonstrates the effectiveness of the proposed von Neumann entropy aware attention mechanism. Disabling the virtual evolution node representation also degrades performance. Furthermore, when the MoTE decoder is removed, the MathOverflow and BitcoinAlpha datasets exhibit more severe drops in performance, demonstrating that the MoTE decoder excels on temporal networks with a long time span and a greater number of evolution states.

### 4.4 Qualitative Analysis of Evolution States

In Fig. 4, we show curves of the von Neumann entropy change for different networks over the same time interval. Compared with the BitcoinOTC dataset, the MathOverflow leads to the generation of more nodes with high degrees, and these have higher entropy. The von Neumann entropy increases at varying rates, depending on the evolution states of the network in question. This reveals the potential relationship between von Neumann entropy and evolution states. Moreover, we can further understand network structure using von Neumann entropy. For evolution states with low entropy, networks tend to be tree-like or string-like and have more low-degree nodes. For networks with evolution states with high entropy there are more high-degree nodes and the networks tend to be fully connected.

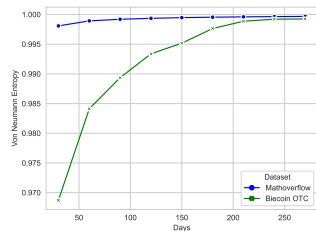

Figure 4: Von Neumann entropy changes on temporal networks.

### 4.5 Parametric Sensitivity

We investigate the sensitivity of our ESSEN to various parameters and evaluate their impact on model performance. In Fig. 5, we report the results of this study and draw the following observations: a) By exploring different training time split points, we evaluate ESSEN's performance with lower training samples and more testing samples. Our model sustains excellent performance even when the number of samples in the training set is reduced. The results clearly show the robustness of ESSEN for complex evolution states. b) Considering the number of experts $Y$, both datasets have "sweet spots". This finding indicates that different datasets exhibit a preference for specific numbers of experts in the MoTE decoder. This can be attributed to the varying complexity of evolution states in the temporal network. c) We observe a strong correlation between the evolution-aware neighborhood size $N$ and the number of nodes in the temporal networks. The link prediction performance decreases when $N \ll |V|$. For example, the total number of nodes in the MathOverflow dataset is 21688, which is also the maximum value of $|V|$ because nodes will added and deleted over time. The AUC results for the MathOverflow dataset reduced from 98.56% to 95.43% when varying $N$ from 200 to 50. This

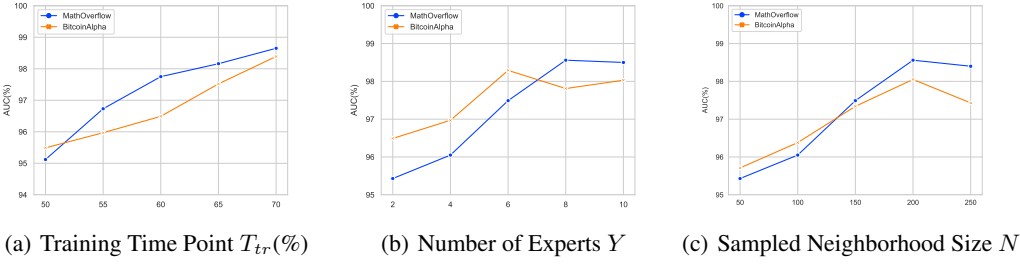

(a) Training Time Point $T_{tr}(\%)$  (b) Number of Experts $Y$  (c) Sampled Neighborhood Size $N$

Figure 5: Study on important settings. We report the results of the inductive link prediction.

result shows a promising trade-off between computational efficiency and the loss of information due to sampling.

## 5 Related Work

**Temporal Network Representation Learning.** Network representation learning is often used to transform large networks into lower-dimensional vectors. For instance, CTDNE [15] learns node embedding from a continuous-time dynamic network instead of a sequence of snapshots. Besides, JODIE [10] uses two recurrent neural networks (RNNs) to learn trajectories of users and items and updates the embedding when interaction occurs. Expanding on this, TGAT [25] utilizes a self-attention mechanism and presents an encoding method to learn inductively. Additionally, CAW [24] and NeuralTWs [6] learn temporal structure using random walk. Specifically, CAW proposes a new anonymization strategy, and NeuralTWs considers structural and tree traversal properties. TDLG[4] aims to model the edges in temporal networks directly instead of calculating from node embedding. Despite these advancements, capturing the global network evolution states within acceptable time complexities remains a formidable challenge.

**Von Neumann Entropy of Static Graph.** Von Neumann graph entropy serves as a pivotal descriptor of a network system's statistical state [3; 16; 17]. Researchers like De Domenico et al. [5] have employed von Neumann graph entropy for structural reduction in multiplex networks, showcasing its versatility. Li et al. [11] explored convergence using von Neumann entropy in the context of network-ensemble comparison, illuminating its potential applications. Moreover, Liu et al. [13] delved into universal patterns of the dynamic genome through von Neumann graph entropy, broadening the scope of its utility. Wang [23] took a different route by approximating von Neumann graph entropy with node degrees, effectively modeling network evolution. Despite these endeavors, the application of von Neumann entropy in representing temporal structures has remained largely unexplored. To the best of our knowledge, our work pioneers the incorporation of von Neumann entropy in temporal network representation learning, opening new avenues for research and exploration in this domain.

## 6 Conclusion

In this paper, we propose ESSEN, an evolution states aware network for recognizing and analyzing the evolution states in temporal networks. We addressed the limitations of existing methods in capturing the time-varying nature of network structures, especially in complex evolution states. Our framework incorporates a von Neumann entropy aware attention mechanism and network evolution state contrastive learning for graph encoding. The decoding stage utilizes a unique decoder referred to as the Mixture of Thermodynamic Experts (MoTE). We evaluated ESSEN on link prediction tasks under transductive and inductive settings and compared it to state-of-the-art baselines. The experimental results demonstrate the effectiveness of our proposed method in capturing temporal dynamics where it outperforms existing approaches. Our work contributes to advancing the field of temporal network analysis and opens up possibilities for future research in both alternative domains and under additional network dynamics. In the future, we will focus on improving ESSEN's efficiency and scalability, allowing it to handle larger datasets and real-time analysis.

## Acknowledgments

This work is supported by the National Natural Science Foundation of China (62176227, U2066213); the Fundamental Research Funds for the Central Universities (20720210047). We would like to thank the anonymous reviewers for their kind comments and valuable suggestions.

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

## A  Additional Algorithm Details

### A.1  The Derivation of Approximate Von Neumann Entropy on Temporal Network

In the following, We commence by summarizing the approximation of the undirected graph von Neumann entropy presented by [27]. First, we introduce the von Neumann entropy, which can be computed from the normalized Laplacian spectrum as follows:

$$S_{VN}(G) = -\operatorname{Tr}(P \log P) = -\sum_{i=1}^{|V|} \frac{\lambda_i}{|V|} \log \frac{\lambda_i}{|V|}, \tag{19}$$

where $\lambda_1, \ldots, \lambda_{|V|}$ are the eigenvalues combinatorial Laplacian matrix. Scaling the normalized Laplacian matrix by the reciprocal of its trace, we obtain a density matrix $\frac{\hat{L}}{|V|}$, The eigenvalues of the density matrix is $\left( \frac{\hat{\lambda}_1}{|V|}, \frac{\hat{\lambda}_2}{|V|}, \ldots, \frac{\hat{\lambda}_{|V|}}{|V|} \right)$ and thus the von Neumann entropy of density matrix associated with the normalized Laplacian matrix of the graph is defined as

$$S_{VN}(G) = -\sum_{j=1}^{|V|} \frac{\hat{\lambda}_j}{|V|} \ln \frac{\hat{\lambda}_j}{|V|}. \tag{20}$$

The von Neumann entropy above relies on the computation of the normalized Laplacian spectrum, therefore its computational complexity is cubic in the number of nodes. The Taylor expansion for $\ln \frac{\hat{\lambda}_i}{|V|}$ is

$$\left( \frac{\hat{\lambda}_j}{|V|} - 1 \right) - \frac{1}{2} \left( \frac{\hat{\lambda}_j}{|V|} - 1 \right)^2 + \frac{1}{3} \left( \frac{\hat{\lambda}_j}{|V|} - 1 \right)^3 - \frac{1}{4} \left( \frac{\hat{\lambda}_j}{|V|} - 1 \right)^4 + \cdots. \tag{21}$$

If we keep the first item of the expansion and discard the remaining that contribute to a small amount, $\ln \frac{\hat{\lambda}_i}{|V|}$ is approximated using $\left( \frac{\hat{\lambda}_j}{|V|} - 1 \right)$. Then the entropy $S_{VN}(G)$ can be replaced by the quadratic entropy $\sum_j \frac{\hat{\lambda}_j}{|V|} \left( 1 - \frac{\hat{\lambda}_j}{|V|} \right)$, then we obtain

$$S_{VN}(G) = -\sum_j \frac{\hat{\lambda}_j}{|V|} \ln \frac{\hat{\lambda}_j}{|V|} \simeq \sum_j \frac{\hat{\lambda}_j}{|V|} \left( 1 - \frac{\hat{\lambda}_j}{|V|} \right)$$
$$= \frac{1}{|V|} \sum_j \lambda_j - \frac{1}{|V|^2} \sum_j \lambda_j^2. \tag{22}$$

Using the fact that $\operatorname{Tr}\left[ \hat{\mathbf{L}}^k \right] = \sum_j \hat{\lambda}_j^k$, the quadratic entropy can be rewritten as

$$S_{VN}(G) = \frac{\operatorname{Tr}[\hat{L}]}{|V|} - \frac{\operatorname{Tr}\left[ \hat{L}^2 \right]}{|V|^2}. \tag{23}$$

The normalized Laplacian matrix $\hat{L}$ has unit diagonal elements. For the trace of the normalized Laplacian matrix we have

$$\operatorname{Tr}\left[ \hat{L} \right] = |V|. \tag{24}$$

Similarly, for the trace of the square of the normalized Laplacian, we have

$$\operatorname{Tr}\left[ \hat{L}^2 \right] = \sum_{u \in V} \sum_{v \in V} \hat{L}_{uv} \hat{L}_{uv} = \sum_{u \in V} \sum_{v \in V} \left( \hat{L}_{uv} \right)^2$$
$$= \sum_{u,v \in V u=v} \left( \hat{L}_{uv} \right)^2 + \sum_{\substack{u,v \in V \\ u \neq v}} \left( \hat{L}_{uv} \right)^2 = |V| + \sum_{(u,v) \in e} \frac{1}{d_u d_v}. \tag{25}$$

Substituting Eq.24 and Eq.25 into Eq.23, the entropy becomes

$$S_{VN}(G) = \frac{\operatorname{Tr}[\hat{L}]}{|V|} - \frac{\operatorname{Tr}\left[ \hat{L}^2 \right]}{|V|^2} = \frac{|V|}{|V|} - \frac{|V|}{|V|^2} - \sum_{(u,v) \in e} \frac{1}{|V|^2 d_u d_v} = 1 - \frac{1}{|V|} - \frac{1}{|V|^2} \sum_{(u,v) \in e} \frac{1}{d_u d_v}. \tag{26}$$

We project the temporal network to the time-independent 2-D plane as an edge-weighted graph, resulting in a simplified depiction of the underlying network structure at a given time. As a result, The expression of the approximate entropy is quadratic in the number of nodes.

Table 3: Performance of AP(%) for link prediction. The best results in each column are highlighted in bold font and the second-best results are underlined.

| Task | Methods | MathOverflow | BitcoinAlpha | BitcoinOtc | Wikipedia |
|------|---------|--------------|--------------|------------|-----------|
| Transductive | JODIE | 84.95 ±0.43 | 90.32 ±0.19 | 91.50 ±0.19 | 92.95 ±2.27 |
| | DyRep | 80.97 ±0.25 | 79.42 ±2.23 | 78.95 ±2.76 | 94.63 ±0.20 |
| | TGN | 81.51 ±1.73 | 86.47 ±0.42 | 88.76 ±1.70 | 98.52 ±0.09 |
| | TGAT | 74.35 ±0.29 | 79.18 ±0.54 | 79.53 ±0.84 | 93.18 ±0.13 |
| | CAW | 61.40 ±0.28 | 71.27 ±0.87 | 79.29 ±0.76 | 98.82 ±0.12 |
| | TDLG | 82.87 ±0.16 | 91.19 ±0.24 | 92.24 ±0.25 | 87.25 ±0.15 |
| | NeurTWs | 93.07 ±0.54 | 94.14 ±0.24 | 96.17 ±0.08 | 96.01 ±0.52 |
| | Ours | **98.60 ±0.26** | **99.06 ±0.20** | **98.83 ±0.47** | **99.04 ±0.26** |
| Inductive | JODIE | 68.58 ±0.49 | 75.02 ±0.20 | 77.44 ±0.14 | 89.33 ±5.04 |
| | DyRep | 65.65 ±0.44 | 66.54 ±1.04 | 65.94 ±0.86 | 91.94 ±0.27 |
| | TGN | 67.04 ±1.42 | 70.52 ±1.06 | 79.74 ±1.21 | 97.83 ±0.16 |
| | TGAT | 62.77 ±0.64 | 67.09 ±0.88 | 68.32 ±1.84 | 94.18 ±0.43 |
| | CAW | 64.79 ±0.31 | 70.70 ±0.93 | 78.21 ±0.29 | 99.11 ±0.13 |
| | TDLG | 70.18 ±2.16 | 79.53 ±3.19 | 80.95 ±6.88 | 53.47 ±2.41 |
| | NeurTWs | 92.68 ±0.40 | 94.16 ±0.27 | 96.44 ±0.34 | 96.12 ±0.22 |
| | Ours | **98.41 ±0.17** | **97.91 ±0.69** | **98.55 ±0.33** | **98.83 ±0.10** |

# B  Additional Experimental Results

## B.1  Performances in Average Precision

Table 3 reports the detailed transductive and inductive link prediction results of AP.

## B.2  Time Comparison

Fig. 6 compares the training times of ESSEN against the second-strongest baseline NeurTWs. For fairness, we use the same batch size for both models and experiment in the same environment. Note that the running time of ESSEN is down quickly because the approximate thermodynamic quantities have been computed at the first epoch and use cache after that.

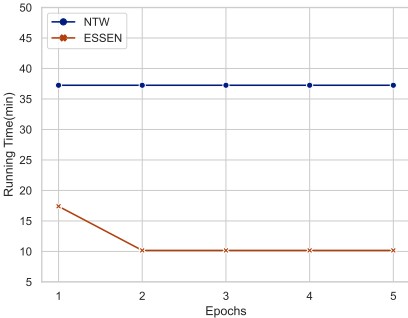

Figure 6: Time Comparison

# C  Experimental Setting

## C.1  Datasets

We introduce the datasets used in this paper as follows. In Fig. 7, we report the degree distribution on the 30th Day and 270th Day in datasets BitcoinOTC and MathOverflow. In order to align the timestamps, we shift the time so that they begin with zero. Additionally, we renumber the nodes to optimize space usage. Further details regarding the preprocessing steps undertaken to prepare these datasets for our method are discussed below.

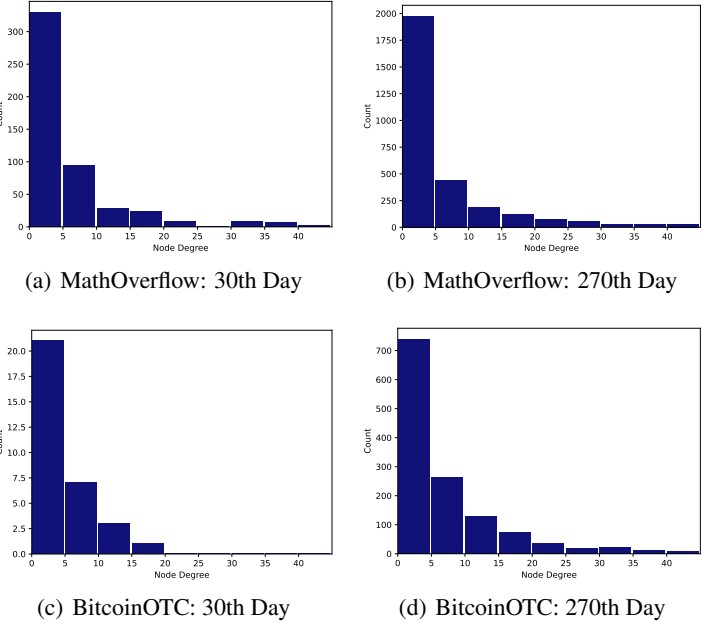

|  |  |
|---|---|
| (a) MathOverflow: 30th Day | (b) MathOverflow: 270th Day |
| (c) BitcoinOTC: 30th Day | (d) BitcoinOTC: 270th Day |

Figure 7: Degree distribution at the 30th Day and 270th Day in different networks.

- **MathOverflow dataset.**[2] It is a temporal network of interactions on the stack exchange website Math Overflow. The nodes represent users, and the edges represent the answers to questions. For example, a directed edge $(u, v, t)$ represents user $u$ answered user $v$'s question at time $t$. Since edge features and node features are not provided, we use the all-zero vectors instead.

- **BitcoinOTC dataset.**[3] It is a who-trusts-whom network of people trade using Bitcoin on the BitcoinOTC platform. Each line records a trade from rater to rate and the rating ranges from -10 to +10 in step 1.

- **Bitcoin Alpha dataset.**[4] It is a similar network to the Bitcoin Alpha platform. Both of the datasets have no edge features or node features, so we initialize them as all-zero vectors.

- **Wikipedia dataset.**[5] It is a dataset of edited records from Wiki pages over a month. We use the top-edited pages and active users as nodes, and each row in our data represents a user editing a page. This dataset records user editing of pages over the course of a month. The timestamp indicates the time when the user edited the page. As with the Reddit dataset, the features of these nodes were processed through LIWC. The user labels indicate if users are temporarily banned from editing.

## C.2 Baselines

The introduction of baselines and their setting details are shown as follows. Baselines not specially mentioned use the default settings of the cited paper.

- **JODIE** uses two recurrent neural networks (RNNs) to learn trajectories of users and items, and updates the embedding when the interaction occurs. we set the number of epochs to 50 and the dimensions of node and time embedding to 100.

- **DyRep** is a temporal point process model capturing both topological evolution and nodes' activities. we set the number of epochs to 50 and the patience for early stopping to 5.

- **TGAT** utilizes a self-attention mechanism and presents a novel encoding method to learn graph embedding inductively. The batch size is 200 and the number of epochs is 50. We set the dimensions of node and time embedding to 100 and 20 neighbors are sampled in aggregation.

---

[2]https://snap.stanford.edu/data/sx-mathoverflow.html

[3]https://snap.stanford.edu/data/soc-sign-bitcoin-otc.html

[4]https://snap.stanford.edu/data/soc-sign-bitcoin-alpha.html

[5]http://snap.stanford.edu/jodie/wikipedia.csv

- **TGN** is a generic and efficient framework for deep learning on dynamic graphs for discrete representation. We set the number of runs to 10 in our experiments. The max number of sampling neighbors is set to 10 and two heads are used in the attention layer. The dropout probability is 0.1 and the learning rate is 0.0001.

- **CAW** utilizes a new anonymization strategy to represent a temporal network inductively. We set the dimension of the positional embedding to 108, batch size to 64, and bias to 1e-5. The maximum number of neighbors when sampling is 64.

- **TDLG** constructs line graphs to model edges directly instead of computing from node embedding. We discard the attributes of the Wikipedia dataset because the original module could not process data with attributes.

- **NeurTWs** improves the causal anonymous walks strategy in CAW and considers structural and tree traversal properties in the process of walking. The dimension of position embedding is 108 and 32 neighbors are sampled for each node. The temporal bias, spatial bias, and ee bias are set to 1e-5, 1, and 0 respectively.

