# OpenReview forum: "ESSEN: Improving Evolution State Estimation for Temporal Networks using Von Neumann Entropy"
_NeurIPS.cc/2023/Conference — NeurIPS 2023 poster_

### Official Review · Reviewer_jRZR · 2023-06-22

**Soundness:** 3 good
**Presentation:** 3 good
**Contribution:** 3 good
**Rating:** 6
**Confidence:** 3

**Summary:**

The authors work on temporal graph representation learning that faces two challenges: (1) the diversity of the evolving patterns and their time-varying nature are hard to model; (2) high computational cost for structure recognization with increasing numbers of nodes and edges.

The authors propose to overcome the problems by incorporating the approximate von Neumann Entropy and approximate thermodynamic temperature difference into the design of temporal graph learning modules. The effectiveness of the proposed method is validated by the link prediction task on different datasets.

Update after rebuttal:
The authors have addressed my concern through the rebuttal. The score remains unchanged.

**Strengths:**

1. This paper is clearly written and easy to follow.
2. This paper brings the thermodynamic view to temporal graph learning, which may inspire later research.
3. The approximate von Neumann Entropy and approximate thermodynamic temperature difference are rigorously formulated and derived.
4. The proposed method significantly promotes the performance of link prediction, and it is computationally efficient.

**Weaknesses:**

The von Neumann entropy and the graph entropy in the thermodynamic temperature difference are defined on undirected graphs. However, the temporal network can be generically directed. How can the proposed method address this issue?

**Questions:**

1. Can the authors provide some intuitive explanations for the quantities $\mathcal{J}(G)$ and $\mathcal{K}(G)$ defined in Eq. (6) and Eq. (7), respectively?
2. In Line 259, the authors say that for large networks, the computational complexity can be reduced from $O(|V|^2)$ to $O(N^2)$, where $|V|$ is the number of nodes, and $N$ is a predefined number to control the budget. Will it make the approximation less meaningful when $N\ll |V|$?
3. In Line 329-331, the authors say that pre-computing some thermodynamic quantities can considerably reduce the computational overhead. However, the network evolves over time, and thus these quantities may be updated. Can the authors explain to what extent the pre-computation can reduce the computational cost?

Comments:
1. Please make the colors of the barplot more distinguishable in Figure 3.
2. It may be inappropriate to place Figure 4 after Figure 5.

---

> ### Author Rebuttal · Authors · 2023-08-09
>
> We appreciate your careful reading, encouraging remarks, and constructive feedback.
>
> **1)	The von Neumann entropy and the graph entropy in the thermodynamic temperature difference are defined on undirected graphs. However, the temporal network can be generically directed. How can the proposed method address this issue?**
>
> It is a great idea.
>
> Temporal networks are abstract representations that are widely used for real-world dynamic systems. In fact, most research on temporal networks focuses on temporal dynamics and simplifies the networks as undirected ones including the state-of-the-art baseline methods (TGAT, TGN, CAW, and NTW). Following these methods, we preprocess the network as undirected for a fair comparison.
>
> Our network can feasibly be covered to a directed one. Specifically, if replacing the undirected degree matrix with an out-degree or in-degree matrix when computing the Laplacian matrix, the computation of von Neumann entropy and the thermodynamic temperature difference can be converted with directionality naturally. Thus, further inclusion of directionality in the study is reasonable and has excellent theoretical potential. We will investigate this deeply in the future.
>
>
> **2)	Can the authors provide some intuitive explanations for the quantities $\mathcal{J}(G)$ and $\mathcal{K}(G)$ defined in Eq. (6) and Eq. (7), respectively?**
>
>   $\mathcal{J}(G)$ and  $\mathcal{K}(G)$ can be explained as the probabilities of a random walker traversing specific edges or cycles in the graph when starting a random walk on the graph. They are the statistics about network structures.
>
> **3)	 In Line 259, the authors say that for large networks, the computational complexity can be reduced from $O(|V|^2)$ to $O(N^2)$, where $|V|$ is the number of nodes, and $N$ is a predefined number to control the budget. Will it make the approximation less meaningful when $N\ll |V|$?**
>
> The link prediction performance decreases when $N\ll |V|$. To quantify the extent of the impact, we provide a parameter sensitivity analysis of $N$ in the manuscript. The total node number of the MathOverflow dataset is 21688, which is also the max value of $|V|$ because the node will add and delete over time. We vary $N$ in the set of \{50,100,150,200,250\}. The AUC results of MathOverflow fluctuate between 95.43% and 98.56%. The result presents a promising trade-off between computational efficiency and the loss of information due to sampling in actual applications.
>
> **4)	 In Line 329-331, the authors say that pre-computing some thermodynamic quantities can considerably reduce the computational overhead. However, the network evolves over time, and thus these quantities may be updated. Can the authors explain to what extent the pre-computation can reduce the computational cost?**
>
> In Line 329-331, the cost which can be reduced is the neighborhood search and the computation of von Neumann entropy in the first epoch of the training. The history structure of the temporal network is fixed. If the adjacency list is sorted by time, it costs $ O(logD)$ to search the neighbors of the node at time $t$, where $D$ is the degree of nodes. Given a test node pair $(u,v,t)$, neighbor search totally costs $O(K^l\times logD)$, where $K$ is the number of neighbors in the neighborhood aggregation process and $l$ is the number of neighborhood aggregation layer. Moreover, it takes $O(N^2)$ in the computation overhead of von Neumann entropy and thermodynamic temperature difference. We hash the test node pair $(u,v,t)$ as key and its neighborhood tree and thermodynamic quantities as value and make these costs reduce to $O(1)$ after the first epoch. After training, the model stores the hash table. If pre-computing the thermodynamic quantities and saving the hash table as the cache files, the training will be faster when loading the hash table directly rather than building it in the first step.
>
> **5)  The comment for figures.**
>
> We will follow the figure suggestion about the colors and position in the revised manuscript.

---

> > ### Comment · Reviewer_jRZR · 2023-08-11
> >
> > Thanks. Your response has addressed my concern.
> >
> > **1)** Your discussion makes things clearer. Since this work is based on literature [27], I believe both the definition of von Neumann entropy and its approximation can be extended to directed graphs.
> >
> > **2)** Please add the explanation to the main text to make the formula more accessible to readers who are less familiar with these quantities.
> >
> > **3)** I have checked Figure 5(c) and understand when the approximation will be practical.
> >
> > **4)** Your complexity analysis and engineering efforts are appreciated.

---

### Official Review · Reviewer_oGt7 · 2023-06-30

**Soundness:** 2 fair
**Presentation:** 2 fair
**Contribution:** 2 fair
**Rating:** 6
**Confidence:** 5

**Summary:**

The paper presents a new framework called ESSEN (Evolution StateS awarE Network) to measure the evolution of temporal networks using von Neumann entropy and thermodynamic temperature difference. Existing methods struggle to handle the time-varying nature of these networks, hindering their performance on complex evolving states. ESSEN utilizes an entropy-aware attention mechanism, contrastive learning, and a unique decoder called MoTE to improve recognition of network evolution states, showing effectiveness in link prediction tasks compared to state-of-the-art methods.

**Strengths:**

Clear motivation and problem statement.
Innovations by integrating domain knowledge in biological study to GNN.


**Weaknesses:**

Some core concepts required domain specific knowledge to fully understand, but these concepts are not clearly defined, such as Von Neumann Entropy.

**Questions:**

What is Von Neumann Entropy?
Why Von Neumann Entropy?

---

> ### Author Rebuttal · Authors · 2023-08-09
>
> Thank you for your careful reading, positive comments, and constructive feedback.
>
> **1)	What is Von Neumann Entropy?**
>
> In the revised manuscript, we will add more details about the definition of von Neumann entropy. Von Neumann entropy is a concept in quantum information theory. In the quantum context, Von Neumann Entropy quantifies the uncertainty associated with the state of a quantum system. This idea has been extended to the static graph domain.
>
> Specifically, the von Neumann entropy is computed from the density matrix for the states of the system under study. the density matrix is used to describe systems whose state is a mixture of pure quantum states $\left|\psi_i\right\rangle$, each with probability $p_i$. The density matrix is defined as
>
> $$
> \rho=\sum_{i=1}^{|{V}|} p_i\left|\psi_i\right\rangle\left\langle\psi_i\right|,
> $$
>  where $|{V}|$ is the number of node set. When defined in this way, the density matrix is Hermitian, i.e., $\rho=\rho \dagger$, $\rho \geq 0$, and Tr[$\rho$] = 1. $\dagger$ represents conjugate transpose. It plays an important role in the quantum observation process, which can be used to calculate the expectation value of measurable quantities. The von Neumann entropy is given by $S_{\textit{VN}}{(G)}=-\operatorname{Tr}(\rho \log \rho)$.
>
> For the graph domain, a density matrix for a graph or network can be obtained by scaling the combinatorial Laplacian matrix $\tilde{L}$ by the reciprocal of the number of nodes in the graph, i.e., $\rho=\frac{\tilde{L}}{|{V}|}$.
>
> The interpretation of the scaled normalized Laplacian as a density operator opens up the possibility of characterizing a graph using the von Neumann entropy. With the definition of the density matrix adopted by Severini et al., the von Neumann entropy can be computed from the normalized Laplacian spectrum as follows:
> $$
> S_{\textit{VN}}{(G)}=-\operatorname{Tr}(\rho \log \rho)=-\sum_{i=1}^{|{V}|} \frac{\hat{\lambda}_i}{|{V}|} \log \frac{\hat{\lambda}_i}{|{V}|},
> $$
> where $\hat{\lambda}\_1$, $\ldots $, $\hat{\lambda}\_{|V|} $ are the eigenvalues combinatorial Laplacian matrix. This form of von Neumann entropy has been shown to be effective for network characterization.
>
> The approximation form of von Neumann entropy also has shown its efficacy for network characterization in the static graph.
>
> **2)	Why Von Neumann Entropy?**
>
> Von Neumann entropy is effective for network characterization. As we discussed in our manuscript, von Neumann entropy is applied to describe the quantum statistics [1] and measure network irregularity [2] in a network system. It offers a novel method to study the properties of pure states and mixed quantum states [3]. Von Neumann entropy measurements play a crucial role in understanding network systems' structural and topological complexity. The ability of von Neumann entropy to capture information content aligns well with the changing nature of temporal networks. This information-theoretic perspective enhances our understanding of the network's evolution laws and capacity to transmit and store information over time.
>
> [1] Passerini, F., Severini, S.: Quantifying complexity in networks: the von Neumann entropy. International Journal of Agent Technologies and Systems (IJATS) 1(4), 58–67 (2009)
>
> [2] Passerini, F., Severini, S.: The von Neumann entropy of networks. Available at SSRN 1382662 (2008)
>
> [3] Anand, K., Bianconi, G., Severini, S.: Shannon and von Neumann entropy of random networks with heterogeneous expected degree. Physical Review E 83(3), 036109 (2011)
>
> **3)	Some Concepts like von Neumann Entropy need to be more clearly defined.**
>
> In the revised manuscript, we will add more details about the entropy concepts and make it more straightforward.

---

### Official Review · Reviewer_w7vF · 2023-07-05

**Soundness:** 2 fair
**Presentation:** 3 good
**Contribution:** 3 good
**Rating:** 6
**Confidence:** 2

**Summary:**

The authors propose a novel method on performing inference tasks on dynamic network structures. Differentiating from previous literature, the proposed method capitalizes on the Von Neumann entropy, which provides a set of indicators about the structural symmetries.

In combination with a quadratic approximation of the Von Neumann entropy, the authors utilize expressions of the thermodynamic temperature differences, in order to create representations of the evolution states of the temporal network. Given the computed representations, the authors propose a decoder based on a mixture of thermodynamic experts, specifically the Von Neumann entropy of the original graph, the Von Neumann entropy of the virtual node graph, and the thermodynamic difference between the two networks.

The experimental study showcases a very strong performance of the proposed ESSEN model, that outperforms (by a large margin in several tasks) the baselines. However, unfortunately I was not able to assess the reproducibility of the results, since no code has been provided until the time of the present review.

**Strengths:**

- The experimental results suggest a very strong performance of the ESSEN model.
- The idea of combining Von Neumann entropy of original and virtual node graph in combination with the thermodynamic difference seems very interesting and can provide some insights on temporal networks.

**Weaknesses:**

- The authors do not provide any clear theoretical indication of the contribution of Von Neumann entropy for representations of dynamic networks.
- It would be really helpful for the community, that by the time of rebuttal the code for the reported results is published.

**Questions:**

- How the approximation of Von Neumann entropy impacts the exact entropy computation? How would be the representations of the dynamic networks would look like given the actual entropy terms?

**Limitations:**

No limitations of the proposed method are discussed. No discussion on potential negative societal impact is discussed.

---

> ### Author Rebuttal · Authors · 2023-08-09
>
> Thank you for the careful reading and helpful comments. Our anonymous code link is provided in the official comment for Area Chairs.
>
> **1)	Theoretical indication of the contribution of Von Neumann entropy for representations of dynamic networks.**
>
> Von Neumann entropy has shown its efficacy for network characterization in the static graph [1]. Our work innovatively introduces von Neumann entropy as a framework for analyzing evolution states. Analyzing evolution states is crucial for the model to better fit network link development law.
>
> The link prediction probability  $P(u,v|G(t))$  is influenced by the network evolution state at time $t$, and the evolution state can vary as the graph's structure changes over time. Von Neumann entropy measures the global uncertainty or randomness associated with a given network evolution state. In the context of link prediction, this uncertainty arises from the various factors that influence whether a link will be established, including past interactions and structural changes in the network. As the network evolves, new information is introduced through the formation of new connections, while existing connections may become obsolete or less relevant. Von Neumann entropy quantifies the information within the temporal network at different time points, shedding light on the information acquisition or loss rate by capturing $\sum_{(u, v) \in E} \frac{1}{d_u d_v}$ in the approximate expression. In the revised manuscript, we will add more details about the theoretical indication of the von Neumann entropy's contribution and make it more straightforward.
>
> [1] : Passerini, F., Severini, S.: Quantifying complexity in networks: the von Neumann entropy. International Journal of Agent Technologies and Systems (IJATS) 1(4), 58–67 (2009)
>
> **2)	How the approximation of von Neumann entropy impacts the exact entropy computation?**
>
> First, the exact von Neumann entropy is defined as
>
> $$
> S_{\textit{VN}}(G)=-\sum_{j=1}^{|V|} \frac{\hat{\lambda}_j}{|V|} \ln \frac{\hat{\lambda}_j}{|V|},
> $$
>
> where $\hat{\lambda}\_{1}$, $\ldots $, $\hat{\lambda}\_{|V|}$ is eigenvalues from normalized Laplacian matrix. And the Taylor expansion for $\ln \frac{\hat{\lambda}_j}{|V|}$ is
>
> $$
> % \begin{split}
>  \left(\frac{\hat{\lambda}_j}{|V|}-1\right)-\frac{1}{2}\left(\frac{\hat{\lambda}_j}{|V|}-1\right)^2+\frac{1}{3}\left(\frac{\hat{\lambda}_j}{|V|}-1\right)^3-
>  \frac{1}{4}\left(\frac{\hat{\lambda}_j}{|V|}-1\right)^4+\cdots .
> % \end{split}
> $$
>
> The key approximation step is keeping the first item of the Taylor expansion for $\ln \frac{\hat{\lambda}_j}{|V|}$ and discarding the remaining that contribute to a small amount. $\ln \hat{\frac{\lambda_j}{|V|}}$ is approximated using $\left(\frac{\hat{\lambda}_j}{|V|}-1\right)$, which holds well when $ \hat{ \frac{\lambda_j}{|V|}} $ is close to 0 or 1. Then we obtain
>
> $$
> \begin{equation}
> \begin{split}
>  S_{\textit{VN}}(G)&=-\sum_j \frac{\hat{\lambda}_j}{|V|} \ln \frac{\hat{\lambda}_j}{|V|} \simeq \sum_j \frac{\hat{\lambda}_j}{|V|}\left(1-\frac{\hat{\lambda}_j}{|V|}\right)
>  =\frac{1}{|V|} \sum_j \hat{\lambda}_j-\frac{1}{|V|^2} \sum_j \hat{\lambda}_j^2 .
> \end{split}
> \end{equation}
> $$
>
> The expression allows us to be expressed in terms of the node degree combinations on edges of the graph in the next steps and by computed with quadratic complexity. In the appendix, we provide the total derivation of approximate von Neumann entropy on the temporal network.
>
> **3)	How would be the representations of the dynamic networks would look like given the actual entropy terms?**
>
> Graphs with low entropy tend to be tree-like or string-like and have more low-degree nodes. Those with high entropy have high-degree nodes and tend to be fully connected. We provide analysis figures about von Neumann entropy and structure evolution process in the attachment PDF.

---

### Author Rebuttal · Authors · 2023-08-09

We sincerely thank all the reviewers for dedicating their valuable time and providing insightful comments. We are greatly pleased to receive some positive reviews. Specifically, we appreciate that they find our work is novel (w7vF), well motivated (oGt7), inspiring (jRZR), well presented (jRZR), and with promising experimental results (w7vF, jRZR). We will incorporate the suggestions and address the concerns in the revision.

To the best of our efforts, we have provided detailed responses to address the concerns raised by each reviewer. Specifically, the primary responses are outlined below:

- We provide the figures of the connection between network structure change and von Neumann entropy.

- We introduce the von Neumann entropy definition more comprehensively.

- We elaborate on the contributions of von Neumann entropy to network representation learning.

- We precisely explain how the approximation process impacts the computation of von Neumann entropy.

- We provide more details of our method, which include providing more insights into thermodynamic parameters $\mathcal{J}(G)$ and $\mathcal{K}(G)$, conducting an in-depth analysis of the relation between $|V|$ and $N$, elucidating the efficacy of pre-computation, and discussing the extensibility for the directed dynamic graph.

Moreover, we would like to emphasize our motivation:

- Von Neumann entropy measurements play a pivotal role in comprehending the structural and topological intricacies of network systems. Its efficacy in network characterization has been well-established. The ability of von Neumann entropy to capture information content aligns well with the changing nature of temporal networks. To the best of our knowledge, research has yet to explore the application of von Neumann entropy in the context of temporal networks. Our work pioneers the extension of approximate von Neumann entropy to temporal networks, aiming to stimulate future research in this area.

- The accurate and adaptive evolution state estimation is paramount for link prediction on temporal networks. By expanding von Neumann entropy to temporal networks, we aim to mitigate the following problems in evolution state estimation: a) Different networks exhibit substantial variations in their evolution laws. Moreover, the evolution states may change over time within the same network. b) Temporal networks accumulate increasing nodes and edges as time progresses, leading to a rapidly expanding neighborhood for each node.

---

### Decision · Program_Chairs · 2023-09-21

**Decision:**

Accept (poster)

**Comment:**

Overall, all three reviewers provide a "Weak Accept" rating for the submission, indicating that it is technically solid with moderate-to-high impact but may have some concerns or areas for improvement. Reviewer w7vF expresses concerns about reproducibility and the theoretical underpinnings of the method, while Reviewer oGt7 notes the need for better clarification of core concepts. Reviewer jRZR raises specific questions about how the method handles directed graphs and computational efficiency, which the authors have addressed in their rebuttal.